# Are the Blueberries We Buy Good Quality? Comparative Study of Berries Purchased from Different Outlets

**DOI:** 10.3390/foods12132621

**Published:** 2023-07-06

**Authors:** M. Teresa Sanchez-Ballesta, Carmen Marti-Anders, M. Dolores Álvarez, M. Isabel Escribano, Carmen Merodio, Irene Romero

**Affiliations:** Department of Characterization, Quality and Safety, Institute of Food Science, Technology and Nutrition (ICTAN-CSIC), Ciudad Universitaria, E-28040 Madrid, Spain; mballesta@ictan.csic.es (M.T.S.-B.); carmarders@gmail.com (C.M.-A.); mayoyes@ictan.csic.es (M.D.Á.); escribano@ictan.csic.es (M.I.E.); merodio@ictan.csic.es (C.M.)

**Keywords:** blueberries, quality, firmness, fungal infection, anthocyanin content, phenolic content, antioxidant capacity

## Abstract

Blueberries (*Vaccinium corymbosum* L.) are becoming increasingly popular for their nutritional and health benefits, and their economic value is therefore increasing. The loss of quality that can occur due to softening and fungal attack is an important consideration when marketing blueberries. Despite the added value of blueberries, no studies have been carried out on how the fruit arrives at the outlets just before purchase by the consumer in terms of firmness, physico-chemical parameters, phenolic compounds, and fungal growth. The aim of this work has been, therefore, to investigate possible differences in quality parameters between blueberries purchased from ten different outlets, regardless of the supplier. The results showed that all the samples were of acceptable quality, although they all had a low maturity index at the point of sale. None of the samples studied showed clear signs of fungal decay at the time of purchase, although we were able to grow and identify some pathogen specimens after cultivation. In terms of total phenolic and anthocyanin content, as well as antioxidant activity, all the samples showed low values, possibly due to their postharvest storage, but they were within the expected range for this fruit. On the other hand, differences in the measured parameters were observed between samples of the same cultivar while no differences were found between conventionally and organically grown blueberries. This suggests that preharvest (such as edaphoclimatic conditions, agricultural practices, and cultivars) and postharvest factors (such as treatments used, storage, and transport temperatures) could influence the berry quality when they reach the consumer.

## 1. Introduction

Blueberry (*Vaccinium* spp.) is a fruit that is highly valued by consumers for its distinctive flavour and aroma. Hailed by the media as a ‘superfood’, due to its high content of health-promoting compounds based on significant amounts of various phytochemicals, blueberries have gained much attention, becoming the second most valued berry in the United States after strawberries (U.S. Department of Agriculture (USDA), Economic Research Service (ERS), 2019). This has led to a 52% increase in global production over the last five years [1]. In this sense, the inclusion of blueberries in the diet is a relatively easy way to add functionality and increase their commercial value.

Among the compounds to which many of the health benefits of blueberries are attributed are phenolic compounds including both flavonoid and non-flavonoid types, the most abundant of which are anthocyanins. They have been shown to have anti-inflammatory, antioxidant, and vasoprotective effects and, with these, a significant modulatory effect on cellular biomarkers related to oxidative stress and inflammation, which leads to chronic diseases such as type 2 diabetes, neurological decline, and cardiovascular disease [2,3]. However, the levels of phenolic compounds in soft fruit are influenced by many factors, including genetic differences, postharvest storage conditions, and the degree of ripeness at harvest [4,5]. In this regard, it has been observed that anthocyanin accumulation in overripe blueberries continues after harvest and during postharvest storage, but it can also decrease depending on the oxidative stress to which it is subjected [6].

An important aspect to consider when marketing blueberries is the loss of quality that can occur due to softening. Fruit softening is a complex horticultural trait that can be caused by numerous factors such as water and turgor loss, cell wall degradation, and membrane damage [7]. In addition, fungal proliferation can also accelerate these processes, as the high levels of sugars and other nutrients and low pH make bacteria less likely to grow, making it easier for yeast and fungi to proliferate. The most common fungi isolated from blueberries belong to the *Botrytis*, *Alternaria*, *Fusarium*, *Penicillium*, *Cladosporium*, *Trichoderma*, and *Aureobasidium* families [8].

Shelf-life is defined as the potential storage time of a product before it becomes unfit for human consumption, or it is rejected by customers, which depends on the quality of the fruit [9]. Thus, quality parameters that are first perceived by the consumer, such as appearance (freshness) and texture (firmness), are very important, but so are others that determine subsequent purchases that are more related to taste and nutritional value, such as soluble solids content (SSC, mainly sugars), titratable acidity (TA), maturity index (MI: SSC/TA), and nutritional quality [10]. Although blueberries are classified as a climacteric fruit, they should be harvested as close to commercial ripeness as possible [8], as they depend on the plant for assimilates and do not improve their organoleptic characteristics, especially sweetness, after harvest, due to the lack of starch reserves. Therefore, blueberries reach their optimum eating quality if they are left on the plant for a few days after they have turned completely blue. The main reason for the perishability of blueberries is their juicy pulp, their high respiration rate, and the fact that they are usually harvested in summer, when temperatures and humidity are high, which increases the respiration rate and thus the ageing metabolism, resulting in weight loss, softening, and decay [11].

Despite the added value of blueberries, no studies have been carried out on how the fruit arrives at retail markets just before purchase by the consumer, in terms of firmness, physico-chemical parameters, phenolic compounds, and fungal growth. Therefore, the aim of this work was to investigate the possible differences in quality parameters, such as firmness, flavonoid content, and fungal contamination, between blueberries purchased on the same day from ten different market places, regardless of the supplier. The results showed differences in the quality parameters analysed. While most of the samples showed quality parameters in accordance with the standards in terms of pH, SSC, and TA, only five of the ten samples showed an adequate SSC/TA ratio. A general characteristic of all of them was the low maturity index reached at the point of sale. On the other hand, samples belonging to the same cultivar also showed differences in terms of the quality parameters between them, suggesting that the growers have probably experimented with different growing conditions and/or storage that affect the quality when they reach the consumer. Finally, no differences were found between conventionally and organically grown blueberries. These results suggest that the loyalty of blueberry consumption to the characteristics of a cultivar is influenced by the place of purchase, which could be of interest for consumers.

## 2. Materials and Methods

### 2.1. Plant Material

Blueberries (*Vaccinium corymbosum* L.) were sampled from ten different retail markets located in Madrid (Spain) in May 2021. Nine samples (1–4, 6–10) were of Spanish origin, three of which belonged to cv. Ventura (1–3), of which sample 1 was grown under organic conditions (Table 1). Sample 4 belonged to cv. Snowchaser and sample number 5, originating from Morocco, belonged to the cv. Royal Blu Aroma. For samples 6–10, the origin is only known to be Spain, but no cultivar information could be obtained. Fruit from each retail market was divided into three lots (biological replicates) and each analysed lot consisted of 136 pooled blueberries. Quality attributes of 40 blueberries from each lot were assessed (titratable acid, soluble solids content, pH, and fungal identification) and the mechanical properties of a further 36 blueberries per lot were analysed. In addition, 60 blueberries were randomly selected, frozen in liquid nitrogen, and stored at −80 °C for further analysis.

### 2.2. Quality Assessments

For titratable acidity analysis, 10 g of the homogenised blueberry sample was diluted with 40 mL of deionised water. TA was measured by titration with 0.1 N NaOH to an endpoint of 8.2 using an 862 Compact Titrosampler (Metrohm, Madrid, Spain) and expressed as % of citric acid (*v*/*w*). The soluble solids content was determined using a Mettler AT100 digital refractometer (Mettler Toledo, Barcelona, Spain). Finally, the pH of the juice obtained from the homogenised blueberries was measured using a micropH 2000 (Crison, Barcelona, Spain). All the measurements were carried out three times. The maturity index was calculated as the ratio of the above parameters (SSC/TA).

### 2.3. Mechanical Properties

The measurements of mechanical properties in blueberries were performed by using a TA.HDPlus texturometer (Stable Micro Systems, Ltd., Godalming, UK) equipped with a 30 kg load cell and with the Texture Exponent Software (6.1.13.0 version). A penetration test was performed by using a 2 mm diameter flat cylindrical stainless-steel probe (P/2). The penetration of the fruits was carried out at the berry equator to a penetration distance of 30% of each berry equatorial diameter. Test speed was set at 0.8 mm/s, considering a trigger test force of 0.1 N (10.2 g). During the penetration test, each blueberry was placed over a flat metal plate with the stem–root axis oriented parallel to the surface. Data acquisition was performed with a frequency of 500 points per second.

From the force–distance curve of each berry, the following mechanical parameters were calculated: the maximum skin-breaking force (N), the distance required to break the skin (mm), the slope of the curve corresponding to skin penetration until break (N/mm) (calculated as the slope of the straight line between the origin and the maximum skin-breaking force), and the work required to break the berry skin (mJ) (calculated as the area under the curve between the origin and the maximum skin-breaking force). The mechanical properties correspond to the average of 36 berries from each of the 10 purchased samples tested.

### 2.4. Fungi Identification

To identify the different pathogens, 10 fresh blueberries were crushed and 1 g was diluted with 9 mL of deionised water. Dilutions were then made from 10–2 to 10–6, from which 100 µL were plated on Petri dishes with Sabouraud Chloramphenicol Agar (SCA) medium (Scharlab, Barcelona, Spain). The plates were incubated for 3 days at 22 °C and the different colonies were isolated and identified using the services of Secugen (Madrid, Spain). DNA was extracted using the DNeasy Plant Pro Kit (Qiagen, Hilden, Germany), and the ITS1-4 region was amplified by PCR using universal primers [12], ITS1: 5′tccgtaggtgaacctgcgg3′ and ITS4: 5′tcctccgcttattgatatgc3′, under the following PCR conditions: 95 °C 15 min: (95 °C 30 s + 53 °C 40 s + 72 °C 1 min) × 35 + 72 °C 8 min. The DNA polymerase used was TaqGold from Applied Biosystems. 

The PCR products were examined by electrophoresis in a 1% agarose gel and visualised using a UV transilluminator. PCR terminators were removed using ExoSap-IT (Applied Biosystems, Waltham, MA, USA). PCR products were sequenced from both ends by Sanger sequencing using Applied Biosystems BigDye 3.1 reagent and then capillary electrophoresed on an ABI 3730xl automated sequencer. The sequences obtained were analysed using Sequencing Analysis software, and the resulting sequences were compared using BLAST with the NCBI database “Internal transcribed spacer region (ITS) from Fungi type and reference material” from NCBI.

### 2.5. Total Phenolic and Total Anthocyanins Content

For the extraction of the total phenolic content, 0.2 g of the pulverised blueberry samples stored at −80 °C were homogenised with 1 mL of a (50:50) solution of methanol acidified (1% HCl)-water (*v*/*v*). The samples were then centrifuged at 10,000× *g* for 10 min at room temperature and the supernatants were collected. These steps were repeated to obtain a final volume of 2 mL. The supernatants were filtered through 0.45 µm nylon filters and stored at −20 °C. The total phenolic content of the extracts was determined by the Folin–Ciocalteu method [13] and expressed as mg of gallic acid equivalents per 100 g of fresh weight (FW).

Total anthocyanin content was determined by the pH differential method [14] with modifications as described by [15]. The results were expressed as mg of cyanidin-3-glucoside (ε = 26.900 L/mol cm) per 100 g FW.

### 2.6. Antioxidant Activity (ABTS and FRAP)

The same extract used for the total phenolic and anthocyanin content was used for the determination of the antioxidant activity. Antioxidant activity was quantified by the ABTS+ method described by [16] and also following the FRAP method [17]. A calibration curve was established using a Trolox solution as a standard reference compound (from 0 to 4 mM). Total antioxidant activity was expressed as µmol Trolox Equivalents (TE) per g FW.

### 2.7. Identification and Quantification of Phenolic Compounds Using HPLC-QTOF

These determinations were carried out on the five samples where the cultivar was known. The same extracts were used for the determination of total phenolic and total anthocyanin contents. Aliquots of the extracted phenolic compounds were analysed using high-resolution chromatography with quadrupole mass spectrometer-time of flight (HPLC-QTOF), as described by [18]. Phenolic peaks were identified through a comparison with standards of chlorogenic acid (C_16_H_18_O_9_), coumaric acid (C_9_H_8_O_3_), caffeic acid (C_9_H_8_O_4_), and quercetin 3-glucoside (C_21_H_20_O_12_) in the range of 0.1 to 100 ppm. For the anthocyanins, the peaks were identified against a standard of malvidin 3-glucoside (C_23_H_25_O_12_), cyanidin 3-glucoside (C_21_H_21_O_11_), cyanidin 3-rutinoside (C_27_H_31_O_15_), delphinidin 3-rutinoside (C_27_H_31_O_16_), and pelargonidin 3-glucoside (C_21_H_21_O_10_) at 1–100 ppm. For the remaining compounds without a standard, identification was based on the presence of identical masses and according to their retention times. The software used was MassHunter Data Acquisition B.05.01 and MassHunter Qualitative Analysis B.07.00.

### 2.8. Statistical Analysis

All the descriptive analyses were performed using the IBM-SPSS statistical program, version 28.0.0 (IBM Corp. Armonk, NY, USA). Differences between blueberries from different markets were determined by a one-way analysis of variance (ANOVA) and the Tukey-b test (*p* < 0.05). Relationships between different analyses were described as Pearson product moment correlation coefficient (r), *p* < 0.01 or *p* < 0.05. 

SPSS also has the ability to perform principal component analysis with quantitative or scalar and qualitative or categorical data (CATPCA). Qualitative or categorical characters were treated as unordered variables (multiple nominal), with the number of character states (categories) entered. All analyses used correlation matrices and two dimensions were extracted to produce scatterplots.

## 3. Results and Discussion

### 3.1. Quality Assessment of Blueberries

Sugar and organic acids have an important influence on the sensory quality of fruit. A good-tasting blueberry should have a high sugar content and a high acidity. Although not all blueberries with high SSC are necessarily good tasting, a low SSC makes it unlikely that they will taste good [19]. According to [20], blueberries should contain more than 10% SSC and have TA values between 0.3 and 1.3% citric acid, an SSC/TA ratio of 10–33, and a pH between 2.25 and 4.25 to be of a good commercial quality. Based on these quality standards, while the ten samples analysed had acceptable TA and pH, only two samples (3 and 6) had an SSC value below 10 and only five (1, 5, 7, 9, and 10) of the ten samples analysed had a balanced SSC/TA ratio (Table 1). These included the two organic samples. It should be noted that the five samples that did not have an adequate SSC/TA ratio did not show large differences in terms of SSC but did show differences in TA, which in some samples was almost half of that observed in the samples that met the quality standards. On the other hand, the five samples with an adequate SSC/TA ratio reached values below 20 in all cases. In the case of samples 1–3 of the Ventura cultivar, only sample 1 of the organic cultivar showed quality parameters according to the standards in terms of SSC/TA ratio. In general, the blueberries analysed from 10 different points of sale showed a low maturity index, considering the recommendations of the quality standards.

### 3.2. Mechanical Parameters

Mechanical properties can be a valuable tool for differentiating the maturity stages of blueberries [21]. The results of this work showed that there were significant differences both in the shape of the force–distance curves obtained from the analysis of the ten samples and in the values of the different mechanical properties derived from them (Table 2). Furthermore, the blueberries with the largest mean value of equatorial diameter (17.6 mm) corresponded to sample 3 and only showed significant differences with samples 1, 2, and 10 (16.0, 15.4 mm, and 15.1 mm, respectively), which were the smallest.

The highest force at skin break (N), which is the force obtained just before the irreversible rupture of the blueberry skin, corresponded to the fruits of samples 7, 8, and 10, although without significant differences with 1 and 9. The lowest values for this parameter corresponded to samples 2–6, although only sample 3, with the lowest value, showed differences with sample 2. 

The distance (mm) travelled by the probe just before skin breakage was greatest in the fruit of sample 1 and was the least in sample 5, with no major differences between the rest of the other samples analysed. The area of the force/deformation curve between the trigger force and the force at the skin break in the blueberries was determined and expressed as skin break work (Table 2). In this case, fruit from samples 1 and 10 had the highest values and, again, sample 5 had the lowest. Similarly, there were no significant differences between the other samples. It is important to note that our retail sampling involves unknown and uncontrolled variables, such as the grower, the supply chain, and the postharvest handling. It is therefore difficult to relate the parameters obtained to the postharvest and/or maturity stage of the fruit. However, we have observed that sample number 5, which presented blueberries with the highest maturity index (SSC/TA:19.03), had the lowest mechanical values described above. In a previous work, [21] reported a decrease in skin breakage force and skin breakage energy as an indicator of the progress of ripening in two different blueberry cultivars (Nui and Rahi). These authors go so far as to conclude that these mechanical parameters could be used for commercial or research purposes, with the aim of being used as quality control operations or to evaluate postharvest technological treatments. Although it is true that, in our study, it was observed that there are samples with different maturity indexes, with half of them below the quality requirements established for these fruits, we were not able to reach the same conclusion in terms of mechanical properties. This may be due to the fact that in our study we used a flat cylindrical stainless-steel probe with a diameter of 2 mm, unlike the needle probe used by [21]. However, other authors have suggested that growing conditions may have a greater influence on the skin penetration test results than the stage of ripening itself [22]. However, unlike the needle probe with a tip diameter of 0.39 mm and a maximum diameter of 2 mm used in the penetration tests by Rivera et al. [21], Mauri et al. [22] performed their experiments with a cylindrical and rounded probe with a smaller diameter of 0.16 mm. Our results indicate that, despite the differences in sugar and acid content found in the different samples, there are no major differences in the mechanical properties analysed. Only sample 5, from the Royal Blu Aroma cultivar from Morocco, showed the highest SSC/TA ratio and the lowest values for the different mechanical properties determined. The fact that it is the only sample with a different origin may indicate that the time between harvest and marketing is longer and that it may have undergone some postharvest treatment. 

### 3.3. Fungi Identification 

Although the samples analysed showed no visible signs of pathogen contamination, it could not be ruled out that the fruits might contain spores of different fungi. A study was therefore carried out to identify them using partial sequencing of the 5.8S rRNA and adjacent intergenic regions. 

Five fungal species were identified in the analysed blueberries (Table 3) corresponding to *Aspergillus tubingensis* or *A. costaricaensis*, *Sporobolomyces roseus*, *Cladosporium piniponderosae* or *C. colombiae*, *Metschnikowia vanudenii* and *Penicillium brevicompactum*. However, not all of them were identified in the ten samples analysed. Thus, none of the five fungi identified were determinate in samples 1, 6, and 8, whereas up to four different fungi were identified in samples 3 and 7. In addition, *P. brevicompactum* was identified in seven samples, and *C. piniponderosae* or *C. colombiae* were found in five of the ten samples analysed, respectively. 

Blueberries, like most fruits, are susceptible to fungal spoilage. Contamination can occur at any stage of the process, from harvesting to consumption, and the more common fungal species may differ depending on the place of production [23]. According to these authors, the most common moulds in blueberries are *Botrytis cinerea* (55%) and *Alternaria* spp. (46%), followed by the *Fusarium*, *Penicillium*, *Aureobasidium*, *Cladosporium*, and *Trichoderma* species. However, *B. cinerea* contamination was not present in any of the samples analysed and *Penicillium* spp. was present in seven samples. A study analysing organic and conventional fruit showed that the fungi that were present in both types of fruit belonged to the genera *Cladosporium*, *Penicillium*, *Alternaria*, and *Aureobasidium* [24]. Moreover, these authors indicated that their presence on the fruit did not depend on the growth conditions of the plant. In this sense, although in our study organic fruit were analysed in two different samples, the results showed that, in one of them, no pathogen was identified, while in the other one, almost three different fungi were identified, including *Cladosporium* spp. This genus was most frequently found in organically grown fruit (45–84% of all fungi identified) in Brigitte Blue blueberries compared with those of the same cultivar grown under conventional conditions [24].

### 3.4. Total Phenolic and Total Anthocyanins Content

The total phenolic content (TPC) measured in the blueberries ranged from 61.94 to 177.94 mg GAE/100 g FW (Figure 1). Sample number 10 showed the highest TPC with 177.94 mg GAE/100 g FW, followed by samples 5 and 7 (without significant differences between them) and, finally, the fruit from samples 1, 3, 6, and 8 showed the lowest TPC values. According to [25], the mean TPC values for blueberries (cv. Bluecrop) at harvest were 274.48 mg GAE/100 g FW, while [26] reported TPC values around 252 mg GAE/100 g FW for the Duke and Bluecrop cultivars, reaching 161 mg GAE/100 g FW in Chandler blueberries. In general, our results showed that the TPC values obtained in this study were lower than those previously reported. It is known that a decrease in the total phenolic content can occur during postharvest storage [27]. 

In terms of total anthocyanin content (TAC) (Figure 2), the highest values were obtained from samples 2 and 7, with 56.66 mg C3G/100 g FW and 56.54 mg C3G/100 g FW, respectively, while the fruit from sample 3 had the lowest value (25.68 mg C3G/100 g FW). In a recent review, Ref. [28] summarised the results of TAC from different blueberry cultivars and different locations, indicating a high variation among blueberry cultivars for total anthocyanin content, although the method of determination was different. The values ranged from 19.3 to 677.8 mg C3G per 100 g FW. In this sense, genotypes and environmental growing conditions could be the main reasons for the differences in total anthocyanin content between cultivars.

It is also important to note that fruit size has an effect on fruit quality. Polyphenols, especially anthocyanins, are mainly found in the skin of blueberries and, for the same weight, smaller blueberries have a greater skin surface area compared with larger fruit. Although it is true that this statement cannot be extrapolated to all the samples analysed, the fruits of sample 10, with the smallest equatorial diameter (Table 2), showed the highest values of total phenolics and the third highest value of anthocyanins. The fruits of sample 3, with the largest equatorial diameter of all the samples analysed, showed the lowest values for both determinations.

In any case, our results show variations in the anthocyanin and phenolic contents among the 10 samples analysed. Finally, it is important to note that any comparison with previous work must take into account all the factors that influence phenolic and anthocyanin content, such as cultivar, postharvest treatments, and rheological and climatic conditions [23], which we do not know in our case.

### 3.5. Antioxidant Activity

The antioxidant activity determined by the ABTS method (Figure 3) showed a wide variation, ranging from 16.65 to 40.96 µmol TE/g FW, corresponding to samples 3 and 5, respectively. Using the FRAP method, the results ranged from 13.22 to 27.38 µmol TE/g FW, corresponding to the samples 6 and 10, respectively. The amplitude of both ranges has been observed in previous studies [25,26,29]. In general, the antioxidant activity values obtained in this study are similar to those reported by [25], of 1014–2055 and 699–1740 µmol TE/100 g FW for the ABTS and FRAP methods, respectively.

The results showed a positive significant correlation between the assessment of antioxidant activity in the blueberry samples using the ABTS and FRAP methods (Table 3) (r = 0.527, *p* < 0.01). A significant correlation (*p* < 0.01) was also found between the antioxidant activity determined by the ABTS or FRAP methods and TPC (ABTS; r = 0.644, FRAP; r = 0.827). Therefore, the presence of phenolic compounds in fruits contributes significantly to their antioxidant activity [30,31]. However, while no correlation was found between TAC and the antioxidant activity determined by the ABTS method, a significant correlation was found in the results obtained by the FRAP method (r = 0.623, *p* < 0.01).

### 3.6. Identification and Quantification of Phenolic Compounds Using HPLC-QTOF 

The identification and quantification of anthocyanins, flavonols, flavanols, and phenolic acids were carried out on samples of known cultivars (samples 1–5). Table 4 shows the main phenolic compounds present in blueberries with their chemical formula, exact molecular weight, retention time (min), and method of identification. The individual anthocyanins were identified by comparing the *m*/*z* of each anthocyanin molecule and its fragmentation with the value in the available published works, as well as through a comparison with standard solutions. The identification of the rest of the phenolic compounds (flavanols, flavonols, and phenolic acids) was carried out with standard solutions and by generating a formula from the MS spectra, generating a similarity score (Appendix A). Twelve peaks of phenolic compounds were tentatively identified in the analysed blueberry samples.

The different anthocyanins were expressed as their pyranoside forms, as the galactoside and glucoside molecules have exactly the same molecular weight and cannot be distinguished by MS/MS. Furthermore, in the work of [32], this distinction was made only by the difference in their retention times. However, in our case, as in [33], some of them were not well-defined peaks and could not be integrated separately. On the other hand, the compounds in the table have been ordered according to the elution time, which is consistent with those found in the bibliography [32,33,34], where the general order is delphinidin, cyanidin, petunidin, peonidin, and malvidin. Quantification was only carried out for those compounds for which standard solutions were available and that were present in sufficient quantities to be measured. The content of individual anthocyanins and chlorogenic acid in blueberries is therefore given in Table 5. 

The only non-anthocyanin phenolic compound that could be quantified was chlorogenic acid, since caffeic, ferulic, ellagic, and gallic acids were below the detection limit. Its content did not differ between the samples, except for sample 5, which had a higher concentration of 59.28 mg/100 g FW, compared with the other samples, which ranged from 30.67 to 35.93 mg/100 g FW. According to [35], the average chlorogenic acid content of blueberries was 131.18 mg/100 g FW, with a minimum of 64.59 and a maximum of 207.50 mg/100 g FW. These values also agree with those reported by [32] of 70 mg/100 g FW. Of the samples analysed, only sample 5 was close to the minimum value recorded, while the other samples showed only half of this content.

For the anthocyanins, quantification was carried out on the arabinoside and pyranoside forms of malvidin, petunidin, and delphinidin, as the cyanidin content was too low to quantify (Table 5). The order of abundance was delphinidin > malvidin > petunidin. This is in agreement with [32,36] who described the predominant anthocyanin class in American blueberries as delphinidin glycosides. The content of malvidin-3-arabinoside varied between all the samples except those of samples 5 and 1, which also showed the highest values. The lowest value was found in sample 3, which was one third lower than the other samples. On the other hand, for malvidin-3-pyranoside, the fruits of samples 3 and 1 showed the lowest values, with 19.78 and 20.03 mg/100 g FW, respectively, while the highest values belonged to sample 2, with 39.91 mg/100 g FW. For the petunidin-3-arabinoside content, sample 2 showed the highest value (9.52 mg/100 g FW), while sample 3 was free of this compound. The same tendency can be observed for petunidin-3-pyranoside, where sample 2 was three times higher than sample 1, which had the lowest value. Finally, the delphinidin derivatives showed the highest content of all anthocyanins. For the rest, sample 2 had the highest content, with 58.07 and 89.65 mg/100 g FW for the arabinoside and pyranoside forms, respectively, followed by sample 4, with 40.62 and 50.50 mg/100 g FW. 

Overall, of the five samples, sample 5 showed the highest level of chlorogenic acid, which is consistent with the fact that it was the sample with the highest TPC value measured using the pH difference method. It should also be noted that when the levels of the individual anthocyanins identified in the five samples were summed, sample 2 showed the highest levels and sample 3 the lowest, as was also the case with the TACs determined by spectrophotometry. The fact that samples 1, 2, and 3 are from the same cultivar and that they differ in anthocyanin levels may be due to preharvest and postharvest practices, which are unknown to us for these marketed fruits. According to [37], cultivation practices are one of the main factors influencing the concentration of anthocyanins in fruits, as well as the different types of diseases affecting the plant, soil and climatic conditions, pest control, and other agronomic factors. On the other hand, the fact that sample 1 was organically grown did not reflect differences with other conventionally grown samples.

### 3.7. Categorical Principal Components Analysis (CATPCA) 

Finally, a CATPCA analysis was performed with the aim of reducing the original set of variables to a smaller set of uncorrelated components that represent most of the information found in the original variables to differentiate the samples. In the CATPCA including all the variables (quality and texture), the first dimension (44.27% of the variability) was positively related mainly to SSC, ABTS, MI, and the mechanical properties displacement at skin break and skin break work. The second dimension (39.02% of the variability) was positively related to the rest of the variables analysed. With respect to dimension 1, all the samples were closely grouped in the middle (Figure 4), with sample 5 being the only outlier. This sample had the highest MI, SCC, and ABTS values and the lowest values for both mechanical properties affecting this dimension. For dimension 2, most of the samples are clustered around −1/+1, with the exception of sample 3, which is below −2 and samples 7 and 10 which are above +1.

## 4. Conclusions

The results of this work showed that the blueberries analysed were of acceptable quality according to the standards, at the point of sale; however, a common general characteristic of the ten samples analysed from different outlets was the low maturity index values. Although the efforts of producers and marketers to maintain the quality of blueberries have focused on maintaining firmness and the absence of rot, this fact must be taken into account, as the maturity index is the main factor responsible for consumer acceptance of the fruit. Furthermore, no differences in CATPCA results were observed between the organic and conventional samples. Finally, we observed differences between samples of the same cultivar depending on the point of sale, reinforcing the idea that the cultivar is not the only factor influencing the quality of the berries.

## Figures and Tables

**Figure 1 foods-12-02621-f001:**
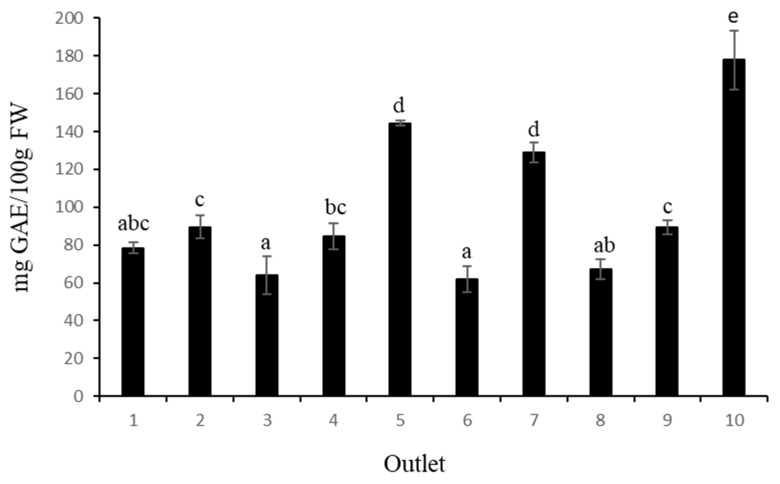
Total phenolic content (TPC) in ten commercial blueberry samples. Values are mean ± SD, *n* = 3. Different letters in the same column indicate that the values are statistically different using the Tukey-b test (*p* < 0.05).

**Figure 2 foods-12-02621-f002:**
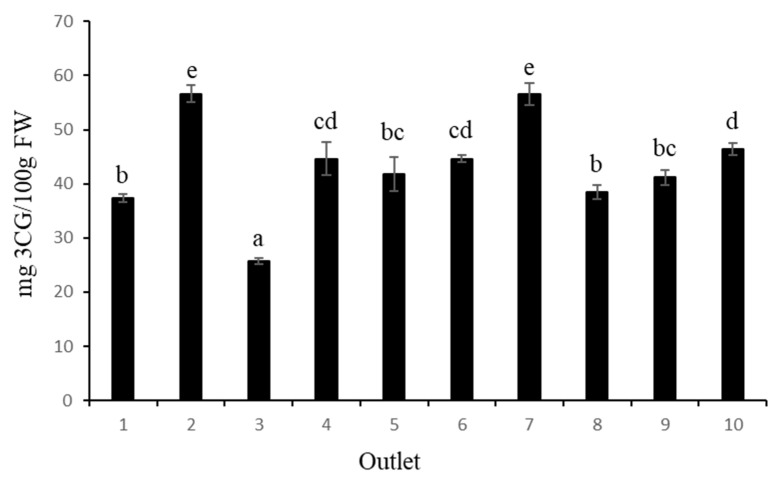
Total anthocyanin content (TAC) in ten commercial blueberry samples. Values are mean ± SD, *n* = 3. Different letters in the same column indicate that the values are statistically different using the Tukey-b test (*p* < 0.05).

**Figure 3 foods-12-02621-f003:**
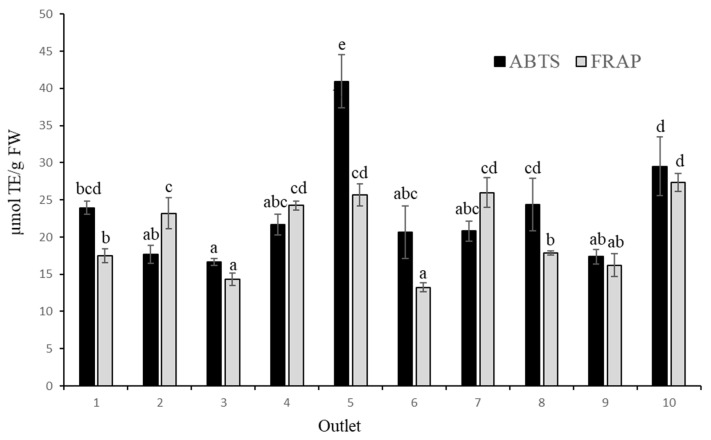
Antioxidant activity, determined by FRAP and ABTS, in ten commercial blueberry samples. Values are mean ± SD, *n* = 3. Different letters in the same column indicate that the values are statistically different using the Tukey-b test (*p* < 0.05).

**Figure 4 foods-12-02621-f004:**
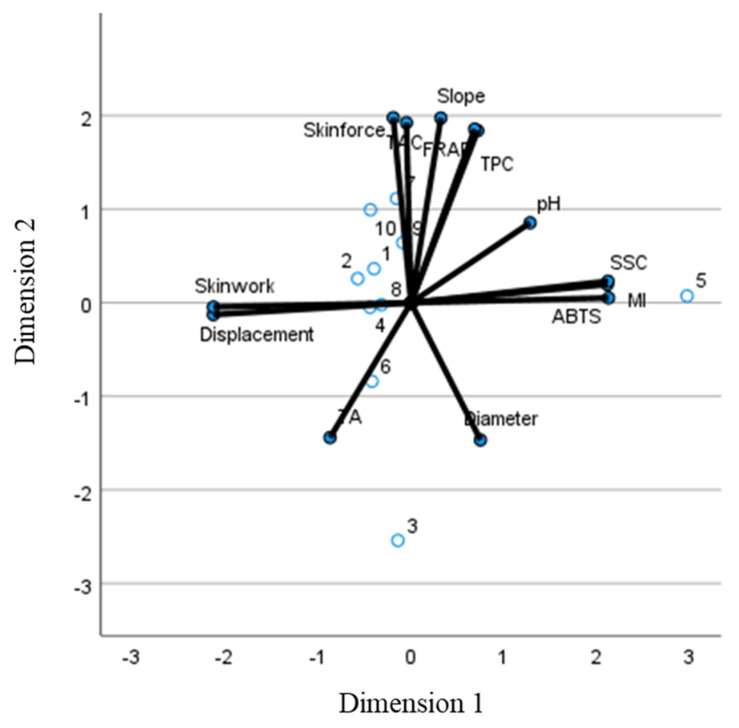
Biplot-CATPCA indicating the observed cases (blue dots: commercial blueberry samples) and component loadings of the variables analysed (black lines).

**Table 1 foods-12-02621-t001:** Blueberry quality parameters obtained from the different outlets.

Sample	TA (% Citric Acid)	SSC (° Brix)	pH	MI (SSC/TA)
1	0.69 ± 0.06 ^a^	11.57 ± 0.23 ^c^	3.39 ± 0.07 ^b,c^	16.84 ^f^
2	1.32 ± 0.02 ^c^	11.27 ± 0.21 ^c^	3.07 ± 0.03 ^a^	8.53 ^a,b^
3	1.30 ± 0.06 ^c^	9.53 ± 0.11 ^a^	3.35 ± 0.09 ^b^	7.35 ^a^
4	1.12 ± 0.01 ^b^	10.10 ± 0.30 ^b^	3.36 ± 0.09 ^b^	9.06 ^a–c^
5	0.66 ± 0.02 ^a^	12.50 ± 0.10 ^d^	3.54 ± 0.03 ^c^	19.03 ^g^
6	1.16 ± 0.04 ^b,c^	9.93 ± 0.06 ^a,b^	3.20 ± 0.02 ^a,b^	8.54 ^a,b^
7	0.77 ± 0.08 ^a^	11.67 ± 0.25 ^d^	3.55 ± 0.03 ^f^	15.15 ^e^
8	1.10 ± 0.09 ^b^	10.33 ± 0.21 ^b^	3.32 ± 0.07 ^b,c^	9.42 ^b,c^
9	1.03 ± 0.06 ^b^	11.03 ± 0.23 ^c^	3.52 ± 0.05 ^d–f^	10.71 ^c,d^
10	1.03 ± 0.12 ^b^	11.77 ± 0.32 ^d^	3.37 ± 0.07 ^b–e^	11.46 ^d^

Different letters in the same column indicate that the means are statistically different using the Tukey-b test (*p* < 0.05).

**Table 2 foods-12-02621-t002:** Mechanical properties of the penetration test corresponding to 10 different commercial blueberries.

Samples	Equatorial Diameter (mm)	Maximum Skin Breaking Force (N)	Displacement at Skin Breaking (mm)	Slope at Skin Breaking (N/mm)	Skin Breaking Work (mJ)
1	16.0 ± 1.8 ^a–c^	1.70 ± 0.37 ^c,d^	2.74 ± 0.64 ^f^	0.63 ± 0.15 ^b^	2.84 ± 1.00 ^d^
2	15.4 ± 1.8 ^a,b^	1.51 ± 0.23 ^b,c^	2.06 ± 0.32 ^b,c^	0.72 ± 0.12 ^b–d^	1.93 ± 0.51 ^a,b^
3	17.6 ± 1.8 ^d^	1.23 ± 0.25 ^a^	2.54 ± 0.48 ^e,f^	0.48 ± 0.11 ^a^	2.01 ± 0.11 ^a,b^
4	16.5 ± 1.9 ^b–d^	1.43 ± 0.29 ^a,b^	2.31 ± 0.36 ^c–e^	0.61 ± 0.13 ^b^	2.12 ± 0.62 ^b,c^
5	16.9 ± 1.1 ^c–d^	1.43 ± 0.25 ^a,b^	1.70 ± 0.34 ^a^	0.83 ± 0.13 ^d–f^	1.58 ± 0.54 ^a^
6	16.8 ± 1.3 ^b–d^	1.42 ± 0.26 ^a,b^	2.08 ± 0.35 ^b,c^	0.67 ± 0.12 ^b,c^	1.90 ± 0.57 ^a,b^
7	16.3 ± 2.3 ^a–d^	1.91 ± 0.46 ^d^	2.20 ± 0.63 ^b–d^	0.89 ± 0.23 ^f^	2.65 ± 1.10 ^c,d^
8	17.4 ± 2.0 ^c–d^	1.87 ± 0.38 ^d^	2.31 ± 0.36 ^c–e^	0.80 ± 0.18 ^d–f^	2.61 ± 0.74 ^c,d^
9	16.6 ± 1.8 ^b–d^	1.68 ± 0.25 ^c,d^	1.94 ± 0.32 ^a,b^	0.87 ± 0.18 ^e,f^	1.94 ± 0.50 ^a,b^
10	15.1 ± 2.2 ^a^	1.85 ± 0.34 ^d^	2.44 ± 0.46 ^d–f^	0.77 ± 0.19 ^c–e^	2.70 ± 0.74 ^d^

Different letters in the same column indicate that the means are statistically different using the Tukey-b test (*p* < 0.05).

**Table 3 foods-12-02621-t003:** Microorganisms identified in the blueberries from different sales outlets.

Microorganisms	Contaminated Samples
*Aspergillus tubingensis* or *A. costaricaensis*	3, 7, 9
*Sporobolomyces roseus*	7
*Cladosporium piniponderosae* or *C. colombiae*	2, 3, 4, 7, 9
*Metschnikowia vanudenii*	3, 4
*Neurospora dictyophora* or *N. tetraspora*	2, 9, 10
*Penicillium brevicompactum*	2, 3, 4, 5, 7, 9, 10

**Table 4 foods-12-02621-t004:** Identification of anthocyanins, flavonols, flavanols, and phenolic acids in individual blueberries in the different samples.

	Compound	Formula	*m*/*z*	T_R_ (min)	Score/Identification
Anthocyanins	Delphinidin 3-pyranoside	C_21_H_21_O_12_	465.1028	9.44	MS/MS
	Delphinidin 3-arabinoside	C_20_H_19_O_11_	435.0922	11.50	MS/MS
	Cyanidin 3-galactoside	C_21_H_21_O_11_	449.1078	11.59	MS/MS
	Petunidin 3-pyranoside	C_22_H_23_O_12_	479.1184	13.16	MS/MS
	Cyanidin 3-arabinoside	C_20_H_19_O_10_	419.0973	15.11	MS/MS
	Malvidin 3-galactoside	C_23_H_25_O_12_	493.1341	16.19	Standard solution
	Peonidin 3-pyranoside	C_22_H_23_O_11_	463.1235	16.39	MS/MS
	Malvidin 3-glucoside	C_23_H_25_O_12_	493.1341	17.01	Standard solution
	Malvidin 3-arabinoside	C_22_H_23_O_11_	463.1235	18.36	MS/MS
Flavonols	Quercetin 3-pyranosides	C_21_H_20_O_12_	464.0949	23.80	Standard solution
Flavanols	Catechin	C_15_H_14_O_6_	291.0863	9.79	97.67
	Epicatechin	C_15_H_14_O_6_	291.0863	14.42	95.79
Phenolic acids	Chlorogenic acid	C_16_H_18_O_9_	355.1024	10.37	Standard solution

**Table 5 foods-12-02621-t005:** Content of phenolic compounds and anthocyanins in blueberry extracts of known cultivars, expressed in mg/100 g FW.

Samples	Chlorogenic Acid	Malvidin 3-Arabinoside	Malvidin 3-Pyranoside	Petunidin 3-Arabinoside	Petunidin 3-Pyranoside	Delphinidin 3-Arabinoside	Delphinidin 3-Pyranoside
1	32.97 ± 2.61 ^a^	30.37 ± 0.91 ^d^	20.03 ± 0.67 ^a^	7.56 ± 0.26 ^c^	5.99 ± 0.31 ^a^	29.14 ± 1.87 ^b^	25.47 ± 1.56 ^a^
2	36.03 ± 0.90 ^a^	23.04 ± 0.39 ^c^	39.91 ± 1.03 ^c^	9.52 ± 0.34 ^d^	19.07 ± 0.83 ^d^	58.07 ± 1.15 ^d^	89.65 ± 3.60 ^d^
3	35.93 ± 2.06 ^a^	8.72 ± 0.96 ^a^	19.78 ± 2.23 ^a^	0.00 ± 0.00 ^a^	9.72 ± 1.32 ^b^	25.99 ± 3.14 ^ab^	40.64 ± 5.84 ^b^
4	30.67 ± 3.71 ^a^	18.22 ± 2.09 ^b^	31.52 ± 3.33 ^b^	6.67 ± 0.74 ^c^	13.53 ± 1.68 ^c^	40.62 ± 2.53 ^c^	58.50 ± 5.91 ^c^
5	59.28 ± 2.29 ^ab^	28.37 ± 1.08 ^d^	30.27 ± 1.25 ^b^	3.44 ± 0.52 ^b^	8.42 ± 0.39 ^b^	22.69 ± 0.17 ^a^	29.07 ± 0.15 ^a^

Different letters in the same column indicate that the means are statistically different using the Tukey-b test (*p* < 0.05).

## Data Availability

Data is contained within the article or Appendix A.

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
