# Peer review of "Are the Blueberries We Buy Good Quality? Comparative Study of Berries Purchased from Different Outlets"

_foods, 2023, doi:10.3390/foods12132621_

Round 1

Reviewer 1 Report

Following your request to review this manuscript, this is my report which contains some remarks. In this paper entitled ' do the blueberries we purchase in different outlets have acceptable quality attributes?. The work is well done, carefully thought out, and performed and the manuscript is well written and falls within the scope of  the Journal . To emphasize the MS for being suitable for consideration in Foods, I have comments as follows: 

  -The seize of writing should respected according to the Journal guidelines, example the Title …… 

-Plant material: please mention plant identifier and whether your specimen has been deposited in a public herbarium under ID

-Fungi identification

  Please note that Molecular identification is needed for Fungi identification

-The authors reported that “Differences between blueberries from dif-ferent markets were determined by a one-way analysis of variance (ANOVA) and the Tukey-b test (p<0.05). “

-Authors mentioned that they used  one-way ANOVA followed by  Tukey-b test,. Did authors check for normality of distributions?. Did the authors check for homogeneity of variances ?.To perform a good statistical analysis, you should check for normality by use of Shapiro–Wilks test and the assumption of homogeneity of variance should be evaluated using Levene’s test. For parameters  normally distributed, a parametric test is required to be used as post hoc test. For parameters that are not normally distributed, the data can be transformed using the natural log (ln) of (x + 1) and then parametric statistics applied or you can apply the appropriate non-parametric statistics.

Authors reported Identification and quantification of phenolic compounds by HPLC-QTOF.”

-  Please where are chromatograms,  so both of analyzed samples and standards should be incorporated,  at least as supplementary material

-  English needs a revision

-   References should be updated to be at least in the last 5 years  

 Moderate editing of English language required

Author Response

We appreciate your valuable comments, which have improved the manuscript. Below you will find the responses to each point in red.

Reviewer 1

Following your request to review this manuscript, this is my report which contains some remarks. In this paper entitled ' do the blueberries we purchase in different outlets have acceptable quality attributes?. The work is well done, carefully thought out, and performed and the manuscript is well written and falls within the scope of the Journal . To emphasize the MS for being suitable for consideration in Foods, I have comments as follows:

-The seize of writing should respected according to the Journal guidelines, example the Title Thank you for the comment.

We have followed Foods Author Guidelines to prepare the manuscript. However, in order to have a more concise Title we have changed it to the following: Are the blueberries we buy good quality? Comparative study of berries purchased from different outlets.

-Plant material: please mention plant identifier and whether your specimen has been deposited in a public herbarium under ID

Thank you for your suggestion. The Accession ID of the blueberries is 6374 according to the herbarium at Aune (http://www.auneherbarium.org/index.html). However, for this type of study it is not necessary to incorporate the specimen in a public herbarium under ID. It was mentioned in the manuscript that commercial blueberries were used. In addition, the cultivar could only be identified for five of the ten samples, as there was no information on the commercial boxes for the rest. We attempted to contact the marketing companies to obtain this information, but the response was unsuccessful and we were unable to include this information. In the case of the Ventura and Snowchaser varieties, they are the result of breeding programs to develop different blueberry varieties that grow well in warm climates. The Ventura blueberry is a product of the nursery genetics program of the US company Fall Creek, while the Snowchaser blueberry was developed by the University of Florida in 2005. Royal Blu Aroma, on the other hand, is the result of research carried out by the Spanish company Royal to develop varieties with a distinctive aroma and flavor.

-Fungi identification Please note that Molecular identification is needed for Fungi identification

Thank you for your comment. We have added more information to improve this section:
2.4. Fungi identification

To identify the different pathogens, 10 fresh blueberries were crushed and 1 g was diluted with 9 mL of deionised water. Dilutions were then made from 10-2 to 10-6, from which 100 μL were plated on Petri dishes with Sabouraud Chloramphenicol Agar (SCA) medium (Scharlab, Spain). The plates were incubated for 3 days at 22ºC and the different colonies were isolated and identified using the services of Secugen (Madrid, Spain). DNA was extracted using the DNeasy Plant Pro Kit (Qiagen), and the ITS1-4 region was amplified by PCR using universal primers (White et al., 1990), ITS1: 5'tccgtaggtgaacctgcgg3' and ITS4: 5'tcctccgcttattgatatgc3', under the following PCR conditions: 95ºC 15 min: (95ºC 30 s + 53ºC 40 s + 72ºC 1 min) x 35 + 72ºC 8 min. The DNA polymerase used was TaqGold from Applied Biosystems. The PCR products were examined by electrophoresis in a 1% agarose gel and visualised using a UV transilluminator. PCR terminators were removed using ExoSap-IT (Applied Biosystems). PCR products were sequenced from both ends by Sanger sequencing using Applied Biosystems BigDye 3.1 reagent and then capillary electrophoresed on an ABI 3730xl automated sequencer. The sequences obtained were analysed using Sequencing Analysis software, and the resulting sequences were compared by BLAST with the NCBI database "Internal transcribed spacer region (ITS) from Fungi type and reference material" from NCBI.”

-The authors reported that “Differences between blueberries from different markets were determined by a one-way analysis of variance (ANOVA) and the Tukey-b test (p<0.05)”.

-Authors mentioned that they used one-way ANOVA followed by Tukey-b test,. Did authors check for normality of distributions?. Did the authors check for homogeneity of variances ?.To perform a good statistical analysis, you should check for normality by use of Shapiro–Wilks test and the assumption of homogeneity of variance should be evaluated using Levene’s test. For parameters normally distributed, a parametric test is required to be used as post hoc test. For parameters that are not normally distributed, the data can be transformed using the natural log (ln) of (x + 1) and then parametric statistics applied or you can apply the appropriate non-parametric statistics.

Thank you for your comment. We checked the normality of the data and the assumption of homogeneity of variance was assessed using Levene's test. The data distribution was normal and p-value of Levene´s test was > 0.05, so we assumed that the variances were homogeneous. ANOVA followed by the Tukey-b parametric test was used to compare the means.

-Authors reported Identification and quantification of phenolic compounds by HPLC-QTOF.”-Please where are chromatograms, so both of analyzed samples and standards should be incorporated, at least as supplementary material.

Thank you for the suggestion. We have added in Supplementary material a file called Supplementary material S1, with chromatograms of standards and identified compounds for the different samples analyzed.

-English needs a revision

We have revised English along the manuscript as Reviewer suggested.

- References should be updated to be at least in the last 5 years.

Thank you for your comment. We have tried to use the most recent studies in the field, but some of the citations are protocols or the first time a paper with results interesting for our study has been published. To solve this problem, we have changed some references (number 4 and former 25, now 26 after adding a new citation in the M&M section). Thus, almost 40% of the total number of references cited in the manuscript are from 2017 to 2023.

4. Horvitz, S. Postharvest Handling of Berries. In Postharvest Handling; Kahramanoglu, I., Ed.; IntechOpen, Ltd.: London, U.K., 2017, doi: 10.5772/intechopen.69073.

26. Cvetković, M.; Kočić, M.; Dabić Zagorac, D.; Ćirić, I.; Natić, M.; Hajder, Đ.; Životić, A.; Fotirić Akšić, M. When Is the Right Moment to Pick Blueberries? Variation in Agronomic and Chemical Properties of Blueberry (Vaccinium corymbosum) Cultivars at Different Harvest Times. Metabolites 2022, 12, 798, doi:10.3390/metabo12090798

Reviewer 2 Report

1.       The purpose of this manuscript is to analyze the various quality attributes of the blueberries that are purchased from local market, but I can non see the scientific value of the work.

2.       The abstract lacks data interpretation to support the discussion, please add more data to abstract.

3.       The abstract concluded that preharvest and postharvest factors could influence the final quality, so I would suggest the authors to state the factors more specifically.

4.       The description of the methods, especially the analytical procedures such as fungi identification, should be referred to published literatures, if possible.

5.       The blueberries in this work were collected from ten different retail markets in Madrid, does this sampling quantity can reflect the common regularity, I would think the sampling numbers are too small to give compelling results.

Author Response

We appreciate your valuable comments, which have improved the manuscript. Below you will find the responses to each point in red.

Reviewer 2

  1. The purpose of this manuscript is to analyze the various quality attributes of the blueberries that are purchased from local market, but I can non see the scientific value of the work.

Thank you for the comment. We have included a sentence in order to highlight the relevance of this work (lines 95-97). Blueberries are increasing in terms of consumption in many countries due to their interesting flavor and healthy properties among other characteristics. In this study we wanted to find out whether the blueberries we buy from different outlets are all of good quality by assessing, among other parameters, their mechanical properties and phytochemical content. Many studies focus on the postharvest quality of the fruit, but there are no studies on the quality of the fruit at the time of purchase. A conclusion of this work is that the ten samples purchased on the same day in large supermarket chains and local shops have low maturity index values, which should be taken into account as it is an attribute linked to consumer acceptance. Thus, in order to ensure firmness or rotting, fruit with a low maturity index is generally sold. On the other hand, no differences were found between organic and conventional fruit. Another conclusion of this study is that the loyalty of blueberry consumption to the characteristics of a variety is influenced by the place of purchase. It is important to develop research that is relevant beyond the scientific community, so we believe these findings are also important for consumers and producers.

  1. The abstract lacks data interpretation to support the discussion, please add more data to abstract.

We agree with Reviewer and the Abstract has been improved in order to support the findings of the manuscript.

Blueberries (Vaccinium corymbosum L.) are becoming increasingly popular for their nutritional and health benefits, and their economic value is therefore increasing. Loss of quality that can occur due to softening and fungal attack is an important consideration when marketing blueberries. Despite the added value of blueberries, no studies have been carried out on how the fruit arrives at the outlets just before purchase by the consumer in terms of firmness, physico-chemical parameters, phenolic compounds and fungal growth. The aim of this work has been, therefore, to investigate possible differences in quality parameters between blueberries purchased from ten different outlets, regardless of the supplier. The results showed that all samples were of acceptable quality, although they all had a low maturity index at the point of sale. None of the samples studied showed clear signs of fungal decay at the time of purchase, although we were able to grow and identify some pathogen specimens after cultivation. In terms of total phenolic and anthocyanin content, as well as antioxidant activity, all samples showed low values, possibly due to their postharvest storage, but within the expected range for this fruit. On the other hand, differences in the measured parameters were observed between samples of the same cultivar while no differences were found between conventionally and organically grown blueberries. This suggests that preharvest (such as edaphoclimatic conditions, agricultural practices and cultivars) and postharvest factors (such as treatments used, storage and transport temperatures) could influence the berry quality when they reach the consumer.

  1. The abstract concluded that preharvest and postharvest factors could influence the final quality, so I would suggest the authors to state the factors more specifically.

We agree with Reviewer and the Abstract has been changed indicating preharvest and postharvest factors that could influence the final quality of the fruit (Lines 28-31).

  1. The description of the methods, especially the analytical procedures such as fungi identification, should be referred to published literatures, if possible.

We agree with Reviewer and we have improved the method of Fungi identification in M&M section as follows:

2.4. Fungi identification

To identify the different pathogens, 10 fresh blueberries were crushed and 1 g was diluted with 9 mL of deionised water. Dilutions were then made from 10-2 to 10-6, from which 100 µL were plated on Petri dishes with Sabouraud Chloramphenicol Agar (SCA) medium (Scharlab, Spain). The plates were incubated for 3 days at 22ºC and the different colonies were isolated and identified using the services of Secugen (Madrid, Spain). DNA was extracted using the DNeasy Plant Pro Kit (Qiagen), and the ITS1-4 region was amplified by PCR using universal primers (White et al., 1990), ITS1: 5'tccgtaggtgaacctgcgg3' and ITS4: 5'tcctccgcttattgatatgc3', under the following PCR conditions: 95ºC 15 min: (95ºC 30 s + 53ºC 40 s + 72ºC 1 min) x 35 + 72ºC 8 min. The DNA polymerase used was TaqGold from Applied Biosystems.

The PCR products were examined by electrophoresis in a 1% agarose gel and visualised using a UV transilluminator. PCR terminators were removed using ExoSap-IT (Applied Biosystems). PCR products were sequenced from both ends by Sanger sequencing using Applied Biosystems BigDye 3.1 reagent and then capillary electrophoresed on an ABI 3730xl automated sequencer. The sequences obtained were analysed using Sequencing Analysis software, and the resulting sequences were compared by BLAST with the NCBI database "Internal transcribed spacer region (ITS) from Fungi type and reference material" from NCBI.”

  1. The blueberries in this work were collected from ten different retail markets in Madrid, does this sampling quantity can reflect the common regularity, I would think the sampling numbers are too small to give compelling results.

Thank you for your comment. The blueberries were bought on the same day from both well-known supermarket chains and local shops representative of the trade in Madrid. While it is true that a larger sample could always give a better idea of the quality of the berries, it is very representative that regardless of the point of purchase or cultivar, the blueberries had a low soluble solids/ titratable acidity ratio, in many cases below the recommended limits. As we have indicated, this fact is extremely important when it comes to accepting the fruit. Although blueberries are classified as a climacteric fruit, they should be harvested as close to commercial maturity as possible as their organoleptic characteristics, especially sweetness, do not improve after harvesting due to lack of starch reserves. These results highlight the difference between the recommended indications for this fruit and what actually happens at the point of sale, which could be relevant for producers and consumers as well as the scientific community.
